

# Eleven years of mountain weather, snow, soil moisture and stream flow data from the rain-snow transition zone - the Johnston Draw catchment, Reynolds Creek Experimental Watershed and Critical Zone Observatory, USA

5 Sarah E. Godsey[1], Danny Marks[2], Patrick R. Kormos[2], Mark S. Seyfried[2], Clarissa L. Enslin[1], Adam H. Winstral[5], James P. McNamara[3], Timothy E. Link[4]

[1] Department of Geosciences, Idaho State University, Pocatello, Idaho, USA
[2] USDA Agricultural Research Services, Boise, Idaho, USA
[3] Department of Geosciences, Boise State University, Boise, Idaho, USA
10 [4] Department of Forest, Rangeland, and Fire Sciences, University of Idaho, Moscow, Idaho, USA
[5] WSL Institute for Snow and Avalanche Research. SLF, Flüelastrasse 11, 7260 Davos Dorf, Switzerland

*Correspondence to*: Sarah E. Godsey (godsey@isu.edu)



**Abstract.** Detailed hydrometeorological data from the rain-to-snow transition zone in mountain regions are limited. As the climate warms, the transition from rain to snow is moving to higher elevations, and these changes are altering the timing of downslope water delivery. To understand how these changes impact hydrological and biological processes in this climatologically sensitive region, detailed observations from the rain-to-snow transition zone are required. We present a complete hydrometeorological dataset for water years 2004 through 2014 for a watershed that spans the rain-to-snow transition zone (doi 10.15482/USDA.ADC/1402076). The Johnston Draw watershed (1.8 km$^2$), ranging from 1497 – 1869 m in elevation, is a sub-watershed of the Reynolds Creek Experimental Watershed (RCEW) in southwestern Idaho, USA. The dataset includes continuous hourly hydrometeorological variables across a 372 m elevation gradient, on north- and south-facing slopes, including air temperature, relative humidity, and snow depth from 11 sites in the watershed. Hourly measurements of incoming shortwave radiation, precipitation, wind speed and direction, and soil moisture and temperature are available at selected stations. The dataset includes hourly stream discharge measured at the watershed outlet. These data provide the scientific community with a unique dataset useful for forcing and validating models and will allow for better representation and understanding of the complex processes that occur in the rain-to-snow transition zone.

**Keywords.** rain-to-snow transition, mixed phase precipitation, watershed hydrology

## 1 Introduction

As the climate warms and many mountain regions shift from snow domination to a mix of rain and snow, we need to understand how these changes will alter hydrologic response [Stewart 2009]. Hydrometeorological conditions in the rain-to-snow transition zone are dynamic with phase changes occurring over short distances and durations [Marks et al., 2013], and while comprehensive datasets are needed, data availability is limited. We present a detailed, serially complete, hourly hydrometeorological dataset from the rain-to-snow transition zone to improve understanding of these complex regions.

The rain-to-snow transition zone in mountainous regions is the elevation band where precipitation phase can vary between rain, snow or a mixture of rain and snow during a single storm event [Marks et al., 2013]. In the northwestern U.S., the elevation of the rain-to-snow transition zone currently ranges from approximately 1500 – 1800 m [Nayak et al. 2010] and covers 9200 km$^2$ [Nolin and Daly, 2006], and varies in both time and space. The extent and elevation of this zone are changing with climate conditions, and vary with latitude and distance from the ocean. The precipitation regime in the current rain-to-snow transition zone in the northwestern U.S. is expected to shift from transitional to a rain-dominated system as the climate warms and the zone moves up in elevation [Nayak et al., 2010; Klos et al., 2014]. Mountain rain-to-snow transition zones are important to study because they are particularly sensitive to changes in climate [Klos et al., 2014]. Because these areas experience winter temperatures near freezing [Mote, 2003; Kormos et al., 2014], small changes in weather conditions





can alter seasonal snow cover, the timing of melt, the delivery of liquid water to soil and streams, and ultimately, the ecosystems they sustain.

Although the rain-to-snow transition zone is recognized as important on regional and continental scales [e.g. Mote et al.,
2005; Klos et al., 2014; Trujillo and Molotch, 2014; Lute et al., 2015], it is surprisingly poorly characterized. Published watershed-scale datasets of precipitation, temperature, humidity, wind, radiation, snow, and resultant streamflow representative of an entire basin spanning the rain-to-snow transition are limited. We conducted a meta-analysis in order to determine published data availability in the rain-to-snow transition zone. We did this by searching the key words: "rain-snow transition data" OR "rain snow zone data" in Web of Science (search date: 4/16/2016). Out of the 79 returns, only 5
publications (6.3%) had published freely available hydrometeorological data in the rain-to-snow transition zone. These 5 datasets are from (1) a small catchment in the Dry Creek Experimental Watershed (DCEW) southwestern Idaho, U.S. [Kormos et al., 2014b], (2) various small- to medium-sized watersheds in the contiguous U.S. [Newman et al., 2015], (3) a site in Washington Cascades, U.S. [Wayand et al., 2015], (4) various sub-watersheds from the Reynolds Creek Experimental Watershed (RCEW) southwestern Idaho [Marks et al., 2013], and (5) a site in Davos, Switzerland [WSL Institute for Snow
and Avalanche Research SLF, 2015]. The remaining 74 returns either were conducted in watersheds that did not span the rain-to-snow transition, or the data associated with their research were neither published nor easily available for public use. It is possible that additional datasets exist, but were not discovered using the search terms that we applied (e.g., data from the H.J. Andrews Long-Term Ecological Research site, Oregon, U.S.A. and a site Col de Porte, France [Morin et al., 2012]). Although data presented by Morin et al. [2012], Wayand et al. [2015], and WSL Institute for Snow and Avalanche Research
SLF [2015] are useful, our dataset is unique because it includes basin-wide measurements and stream discharge, which permit hydrologic modelling and a mass balance approach to validation using soil moisture and streamflow records.

In this paper, we present a comprehensive hydrometeorological dataset for 11 water years (WY, 1 October through 30 September) from WY 2004 – 2014 for the Johnston Draw (JD) watershed that spans the rain-to-snow transition zone in
southwestern Idaho. The dataset is unique not only because the site falls within this climatically sensitive zone, but also because it has instrumentation that encompasses information on the effects of both elevation and aspect on snow accumulation, and soil moisture and temperature. The dataset includes measurements of soil temperature and moisture that support studies of the interactions between the atmosphere and the ground surface. The dataset spans a time period in which conditions were warmer than previous years of record, possibly representing what can be expected as regional climate
warming advances. Our objective is to provide this high spatiotemporal resolution dataset to study short-term variations at intra-event, intra-annual, and inter-annual scales, and we plan to continue these observations to assess long-term climatic trends at the sensitive rain-to-snow transition zone.



## 2 Site Description

The Johnston Draw (JD) is a 1.79 km$^2$ sub-watershed of RCEW and is located in southwestern Idaho (Figure 1). RCEW is managed by the U.S. Department of Agriculture (USDA) Agricultural Research Service (ARS) Northwest Watershed Research Center (NWRC) and is also a National Science Foundation Critical Zone Observatory (CZO). The elevation at JD

ranges between 1497 – 1869 m, spanning the rain-to-snow transition zone where the precipitation phase at higher elevations is snow-dominated, and rain-dominated at lower elevations. Over the period of record, magnitude-weighted incoming precipitation was 39% and 53% snow at the lowest and highest precipitation gages, respectively. The annual average air temperature is 8.1°C, with precipitation averaging 609 mm annually, based on all measurement sites (also see Table 1). Vegetation on the northern-facing slopes is characterized by snowberry (*Symphoricarpos*), big sagebrush (*Artemisia*

*tridentate)*, aspen (*Populus tremuloides)* groves and low sagebrush (*Artemisia arbuscula*) with wheatgrass (*Elymus trachycaulus)* while southern-facing slope vegetation includes *Artemisia arbuscula*, *Elymus trachycaulus*, mountain mahogany (*Cercocarpus ledifolius*), and bitterbrush (*Purshia tridentate)* [Stephenson, 1970]. The dominant soil texture on both north- and south-facing slopes is classified as sandy loam and soils are shallower (~ 50 cm deep) on south-facing slopes compared to north-facing slopes (~ 100 cm deep) [USDA, 2015]. The bedrock in the watershed consists mainly of granitic

rock (79%), with some basalt (3%), and welded tuff (18%) [Stephenson, 1970].

The dataset includes data from 11 meteorological stations and one streamflow station (Table 1). Three full meteorological stations (124, 124b, and 125) measure an extensive suite of variables, including air temperature, relative humidity, wind speed and direction, incoming shortwave radiation, precipitation, snow depth, and soil temperature and moisture. The additional 8 meteorological stations (jdt1, jdt2, jdt3, jdt4, jdt5, jdt2b, jdt3b, and jdt4b) measure select variables for specific

purposes (see Table 1 and below for details). In 2002, the ARS installed full meteorological stations at the bottom (site 125) and top (site 124) of JD, and a weir at the outflow (site 125b). During 2003 – 2005, an additional 5 meteorological stations (jdt1, jdt2, jdt3, jdt4, and jdt5) were installed on the north-facing side of JD to provide measurements of air temperature ($T_a$), humidity (RH), wind speed and direction ($w_s$ and $w_d$, respectively) and snow depth ($z_s$) – so that along with stations 125 at the bottom and 124 at the top – a measurement site was established for every 50 m of elevation in the JD catchment. In 2005,

a full meteorological station (site 124b) was established in an aspen grove near the top of JD to provide weather data at a wind-sheltered site. In 2010, three additional stations (jdt2b, jdt3b, and jdt4b) were installed on the south-facing side of JD, at roughly the same elevations as jdt2, jdt3 and jdt4 on the north-facing side (Figure 1). At the same time, instruments to measure soil temperature ($T_g$) and moisture ($\theta$) were added to 8 of the 11 sites (jdt1, jdt2, jdt3, jdt4, jdt2b, jdt3b, jdt4b, and 124b) starting at 5 cm depth below the surface and then every ~15 cm to 50 cm depth. Sensors were also placed at 75, 90 and

100 cm below the ground surface wherever possible. The maximum depth at each site depends on the depth to bedrock because the instruments could not be installed in bedrock or saprolite. Two soil profiles were installed at 124b due to large vegetation differences within a small area: 124ba, which is located in an aspen grove, and 124bs, which is located in



mountain sagebrush. Details of the sensors used to measure each parameter, as well as the sensor accuracy, operating range, and temperature dependence are provided with the data.

## 3 Data Description

### 3.1 Meteorological Data

All data presented here were checked for time inconsistencies based on the World Meteorological Organization's QA/QC standards [Zahumenský, 2004], using the plausible instantaneous value ranges and maximum/minimum step changes outlined therein. The data were corrected and gap-filled using linear interpolation for gaps less than 3 hours or multiple linear regression for longer gaps from published measurements of the same variable at nearby long-term Reynolds Creek stations 144 and 145. Because additional sites were added during the period of record, sometimes gaps were filled by

different neighbouring sites during different periods. All observations were recorded on an hourly time step with varying start dates for each station (Table 1). We have condensed the relatively large amount of data into summaries to convey conditions within the watershed. For this purpose, we chose two representative WYs; WY2011 was a cool and wet year, and WY2014, a warm and dry year. These WYs were selected to illustrate subsequent figures and analyses because we assumed these two years represent the range and diversity of conditions during the 11-WY time period.

### 3.1.1 Temperature and Relative Humidity

$T_a$ and RH were measured continuously at all sites in JD from WY2004 – WY2014. Water vapor pressure ($e_a$) and dew point temperature ($T_d$) were calculated using measured $T_a$, RH, and software tools from the Image Processing Workbench (IPW) [Frew, 1990; Marks et al., 1999b]. The IPW tools are optimized for temperatures near freezing (0°C) providing greater accuracy for $e_a$ and $T_d$ as $T_a$ approaches 0°C. This accuracy is critical for the determination of precipitation phase in the rain-

to-snow transition zone.

We define a storm as a period of time during which there are no more than 2 consecutive hours without measurable precipitation. $T_a$ during storms is 6.6°C cooler than $T_a$ during non-storms, reflecting seasonal regional precipitation patterns and the dominance of winter storm events, whereas $T_d$ during storms is 1.7°C warmer than $T_d$ during non-storms. Figure 2 shows the average monthly temperatures for $T_a$ and $T_d$ for non-storms (panels a and c) and storms (panels b and d) in

WY2011 and WY2014. For both non-storm and storm periods, the mean $T_a$ and $T_d$ are also close to 0°C for roughly 8 months out of the year (October – May), whereas during summer months (June – September), these temperatures are significantly warmer than 0°C. These mean values demonstrate the sensitivity of JD to climate warming, as changes in temperature and humidity are likely to strongly impact precipitation phase at this location (Nayak et al. 2015).





### 3.1.2 Radiation

Incoming shortwave radiation (280-2800 nm) ($S_i$) was measured continuously at three elevations (stations 125, 124b, and 124). Station 124 $S_i$ had to be occasionally gap-filled because ~ 0.5% of the time series was missing. Only nighttime hours were missing, so gaps were replaced with zeros, matching the other two stations. The WY averages for sites 125, 124b, and

124 are 172.5, 195.6, and 193.6 W m$^{-2}$, which equates to 14.9, 16.9, and 16.7 MJ m$^{-2}$ day$^{-1}$. Peak daily incoming shortwave radiation occurs over a much broader period during the summer: peak $S_i$ typically occurs ~10:00 am – 4:00 pm (Mountain Standard Time (MST)) during summer, and ~12:00 pm – 2:00 pm (MST) during winter.

Longwave radiation is important in many energy balance applications, such as simulating snowmelt and evapotranspiration

[Flerchinger et al., 2009], and Raleigh et al. [2016] show that lack of longwave radiation data can limit model performance. However, many measurement networks, including JD, lack instrumentation to measure this variable and it is thus not reported in our dataset. Nonetheless, longwave radiation is measured within RCEW at a slightly higher elevation (2034 m) at site 176 approximately 3 km to the SE of JD [Marks et al., *data set in preparation*]. Alternately, clear-sky longwave radiation can be accurately calculated [Flerchinger et al. 2009] based on $T_a$, $e_a$, or precipitable water, using methods by

Ångström [1918], Prata [1996], or Dilley and O'Brien [1998].

### 3.1.3 Wind

Wind speed ($w_s$) and direction ($w_d$) were continuously measured at seven sites. Ranging from 0.4 (the instrument threshold) to 24.2 m s$^{-1}$ for all sites, $w_s$ is greatest at 124 because this site is heavily exposed to wind (Table 1). In fact, $w_s$ at the exposed 124 site is on average twice that of all other sites at 4.5 m s$^{-1}$ compared to 2.8 m s$^{-1}$ for all the sites. During storms,

wind speeds are on average 1.4 times faster than during non-storms. During winter storms, $w_d$ ranges from 180 to 220° (measured clockwise from north) whereas $w_d$ usually ranges between 135 and 225°. These values agree with the relatively consistent wind directions of 175 to 230° observed in other sub-watersheds of RCEW and in the nearby DCEW [Winstral et al., 2013; Kormos et al., 2014].

### 3.1.4 Precipitation

The dataset includes wind-corrected ($ppt_a$) precipitation measurements for three sites in JD (125, 124, 124b) and the percentage of precipitation that is in the form of rain, snow, or a mixture of rain and snow that was calculated using the humidity-based methods developed by Marks et al. [1999, 2013]. The precipitation data for stations 125 and 124 were wind-corrected using the dual-gage correction methods developed by the ARS [Hanson et al., 2004], which incorporate a conventional under-catch correction [Hamon, 1973]. Because the 124b site has only a single gage, the dual gage correction

methods cannot be applied to this site. Instead the shielded data for 124b was wind-corrected using WMO [2008] methods. Wind exposure at the upper measurement site 124 results in roughly the same precipitation as at the lower elevation site 125.



Precipitation catch at the sheltered site 124b is on average 1.2 times greater than at the wind-exposed site 124 (Table 1). Based on water balance methods, we believe that the wind-exposed values are too low and that measurements are more representative at the sheltered sites. Thus, we suggest that an orographic lapse rate using only sites 125 and 124b better represents the true precipitation lapse rate. We approximated precipitation for site 124b for WYs 2004 – 2007 via multiple
linear regression using nearby precipitation measurement sites, which were within 1 km horizontally and within 100 m of the same elevation. Mean cumulative $ppt_a$ for the 11 WYs for stations 124, 124b and 125 was 563, 700, and 564 mm, respectively (Table 1).

### 3.2 Stream, Snow, and Soil Data

### 3.2.1 Stream Discharge

Stream discharge was measured continuously with a stage recorder using a drop box weir at the watershed outlet [Pierson et al., 2001]. The intermittent stream draining JD typically starts flowing in early November as winter seasonal precipitation resumes and ceases to flow around mid-July. Stage height was converted to stream discharge using a rating curve [Pierson and Cram, 1998] and frequent field measurements to ensure high-quality flow records [Pierson et al., 2001]. Average stream discharge over the period of record is approximately 0.007 $m^3$ $s^{-1}$ with the largest discharge of 1.63 $m^3$ $s^{-1}$ on 14 February
2014 during a rain-on-snow event. Total annual runoff for each WY is shown in Figure 3.

### 3.2.2 Snow Depths

Instantaneous snow depths were collected at all 11 sites on an hourly basis for all periods when each Judd Communications depth sensor was installed. Raw snow depths from all stations were processed in a multi-step fashion analogous to methods evaluated by Ryan et al. (2008). We first defined the start and end of the snow-covered period for each WY, the peak snow
depth, and a smoothing window for each sensor (usually 8 hours, but under specific circumstances extended to 40 hours as detailed below). Because JD snow cover is often ephemeral, the start of the snow-covered period was defined as the first day with a positive snow depth after the start of the new water year, and the end of the snow-covered period was the last day with positive snow depth during that water year. Thus, the snow-covered period may include periods without snow cover if ephemeral snowpacks melted, especially during the fall and spring. Furthermore, because the snow depth sensor is unreliable
during storms due to the ultrasonic signal reflecting from hydrometeors, these values were filtered and removed. If the gaps that this created were longer than the specified smoothing window, they were not filled. There were 127 unfilled gaps for all stations and years. If gaps were shorter than the smoothing window, then missing data were interpolated. This smoothed dataset was further quality-checked by visually comparing cumulative precipitation and changes in snow depth. If snow depth increased while precipitation was zero, we extended the typical 8-hour smoothing window to 40 hours to minimize
incorrectly interpreting noise as the snow depth signal. Thus, if snow depth decreased during a storm due to compaction, these data were smoothed and preserved. Mean snow depths can be found in Table 1. As expected, north-facing slopes and




sheltered sites have deeper snowpacks that last longer throughout the snow season compared to south-facing slopes and wind-exposed sites (Figure 3), primarily due to shortwave radiation and scour differences.

Although this dataset does not include snow water equivalent (SWE) measurements, which complement snow depths, it appears likely that methods of converting LiDAR-derived snow depth to SWE may soon allow conversion of the 11

continuous snow depth measurements [Kirchner et al., 2014] to SWE, and some snow models (e.g., SNOWPACK (Lehning et al. 1999)) can utilize snow depth measurements to simulate SWE as part of avalanche hazard assessment. We expect that an improved understanding of snowmelt and soil frost may build on these observed snow depth, and soil moisture and temperature measurements.

### 3.2.1 Soil Moisture and Temperature

Soil moisture probes were installed at 8 of the 11 sites at various depths (Table 1) in 2010 to measure soil temperature ($T_g$) and moisture ($\theta$). Mean WY soil temperatures reflect distinct aspect differences (Figure 4) with mean soil temperatures of 7.7°C on north-facing slopes and 12.2°C on south-facing slopes at a depth of 20 cm.

Processing of the soil moisture data included correcting extremely dry measurements resulting from sensors with bad components. The faulty equipment was not immediately apparent because errors are only expressed when water contents are

very low. Thus, the reported values are accurate for all the hydrologically active periods. During the summer dry down and winter freezing events, once a value of about 0.08 $m^3$ $m^{-3}$ is reached, the data drop rapidly to unrealistic values, and when water contents rise due to precipitation inputs or thawing, they return to accurate values. In order to make a continuous estimate of water content and storage, we replaced the faulty values using continuous values from adjacent functional sensors. From these corrected values, we calculated the average water storage (Figure 4) for the north-facing slopes using a

soil depth of 100 cm and for south-facing slopes using a depth of 50 cm, based on the typical depths to which the sensors could be installed. For both WYs, water storage on north-facing slopes is on average 1.4 times greater and lasts longer than on south-facing slopes.

### 4 Mass Balance Analysis

We estimated evapotranspiration using a mass balance approach based on measured precipitation and stream discharge to

ensure that the measurements were reasonable when integrated at a watershed scale. We assumed that net groundwater fluxes and annual changes in storage were negligible, which is supported by soil moisture data. We approximated the annual precipitation of JD by arithmetically averaging the annual precipitation from 125 and 124b (as described in section 3.1.4, site 124 is located on a very wind exposed ridge, and is therefore not representative of the general precipitation lapse rate in the catchment). WY precipitation was estimated to be 765 ± 78, and 548 ± 69 mm year$^{-1}$ for WY2011 and 2014, respectively,

with uncertainties based on the reported instrument error. We converted annual stream discharge to specific discharge by normalizing the annual stream discharge to the watershed area (181.35 ha). These values were estimated to be 309 ± 37 and 89 ± 11 mm year$^{-1}$ for WY2011 and 2014, respectively, assuming up to a 12% annual discharge uncertainty. Uncertainty in



the individual discharge or precipitation measurements was propagated to the annual discharge and precipitation estimates via a simple sum or average (following Morgan and Henrion 1990). We then used the hydrological mass balance equation to estimate evapotranspiration (ET = P – Q + Δstorage), which was approximated to be $456 \pm 86$ and $459 \pm 69$ mm year$^{-1}$ for WY2011 and WY2014, respectively. These estimates agree closely with the value of 425 mm year$^{-1}$ measured in the nearby

Reynolds Mountain East catchment using eddy covariance techniques [Flerchinger et al., 2010].

## 5 Data Availability

All data presented in this paper are available from the USDA National Agricultural Library (see Godsey et al., 2017, https://doi.org/10.15482/usda.adc/1402076). The directory includes a readme file in PDF format listing the contents within each directory with a detailed data description, naming conventions, instruments used, and contact information for additional

inquiries, a watershed digital elevation model, and shapefiles of the watershed boundary and the station locations. The readme file defines the formats for the 3 precipitation files, 11 meteorological data files, 1 stream discharge file, 1 snow depth file, and 8 soil temperature and moisture files. Header descriptions in the associated files indicate subscripts used throughout this manuscript with the _ symbol appearing before subscripted characters.

## 6 Conclusions

The dataset presented is the most complete and comprehensive available to date from the rain-snow transition zone. It includes 11 water years (2004 – 2014) of continuous hourly meteorological data including air and dew point temperature, relative humidity, vapor pressure, precipitation, wind speed and direction, and shortwave radiation at 50 m elevation intervals spanning the JD catchment. Other data include snow depth, stream discharge, and soil moisture and temperature. The dataset is important for a variety of scientific questions because it: (1) captures complex atmosphere-surface-subsurface

dynamics in the rain-to-snow transition zone, (2) represents hydrometeorological differences along both an elevational gradient and between aspects, and (3) provides all the necessary data required for applying a variety of models. It is our intention that this dataset will be used by scientists to improve understanding of the basin-scale interactions and responses for a mountain watershed transitioning from snow- to rain-dominated. High-resolution hydrometeorological datasets can offer researchers opportunities for interdisciplinary studies at the watershed scale. For example, future studies might leverage

these data to better understand how (1) changes in precipitation magnitude and temperature impact water storage and movement, and the average annual snowline, (2) soil temperature and moisture respond to changes in energy and water fluxes, and (3) changes in climate impact subniveal biogeochemistry beneath transient snowpacks at the rain-snow transition.

## 7  Copyright statement



## 8 Supplement link:

No supplements are associated with this article.

## 9 Author contribution

S. Godsey, D. Marks, P. Kormos, M. Seyfried, and C. Enslin prepared the datatset. Datasets were collected by D. Marks, M.
Seyfried, T. Link, J. McNamara, A. Winstral and ARS staff. C. Enslin, S. Godsey, and D. Marks prepared the manuscript
with contributions from all co-authors.

## 10 Acknowledgments

The data presented in this paper were collected by the USDA NWRC. Adam Winstral and Mark Seyfried designed the
expanded JD measuring network. We also thank the NWRC and the Idaho State University Department of Geosciences for
support. The collection and processing of the data presented in this paper were funded in part by the NSF-EPSCoR Program
(IIA-1329469), NSF-CBET (0854553), USDA-ARS CRIS Snow and Hydrologic Processes in the Intermountain West
(5362-13610-008-00D), USDA-NRCS National Water and Climate Center-Portland, Oregon (60-5362-4-003), and the NSF
Reynolds Creek CZO Project (58-5832-4-004).

## 11 Disclaimer

Any reference to specific equipment types or manufacturers is for informational purposes and does not represent a product
endorsement. ISU and USDA are equal opportunity providers.

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

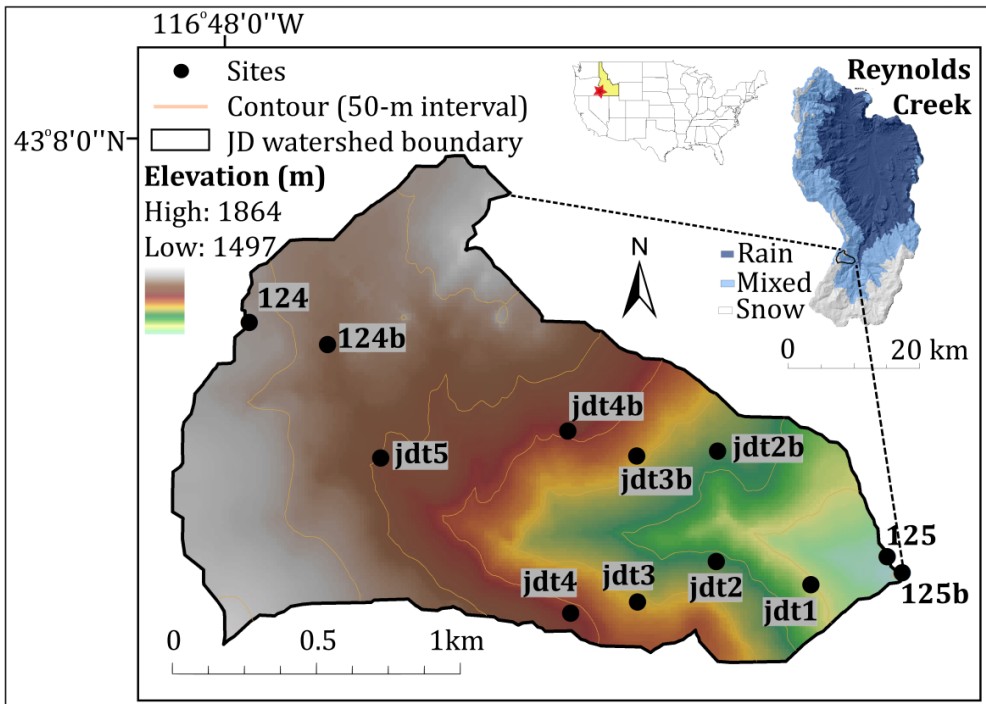

**Figure 1: Johnston Draw (JD) with instrumentation location. For more information on each site, refer to Table 1 and the naming convention file attached to the data.**





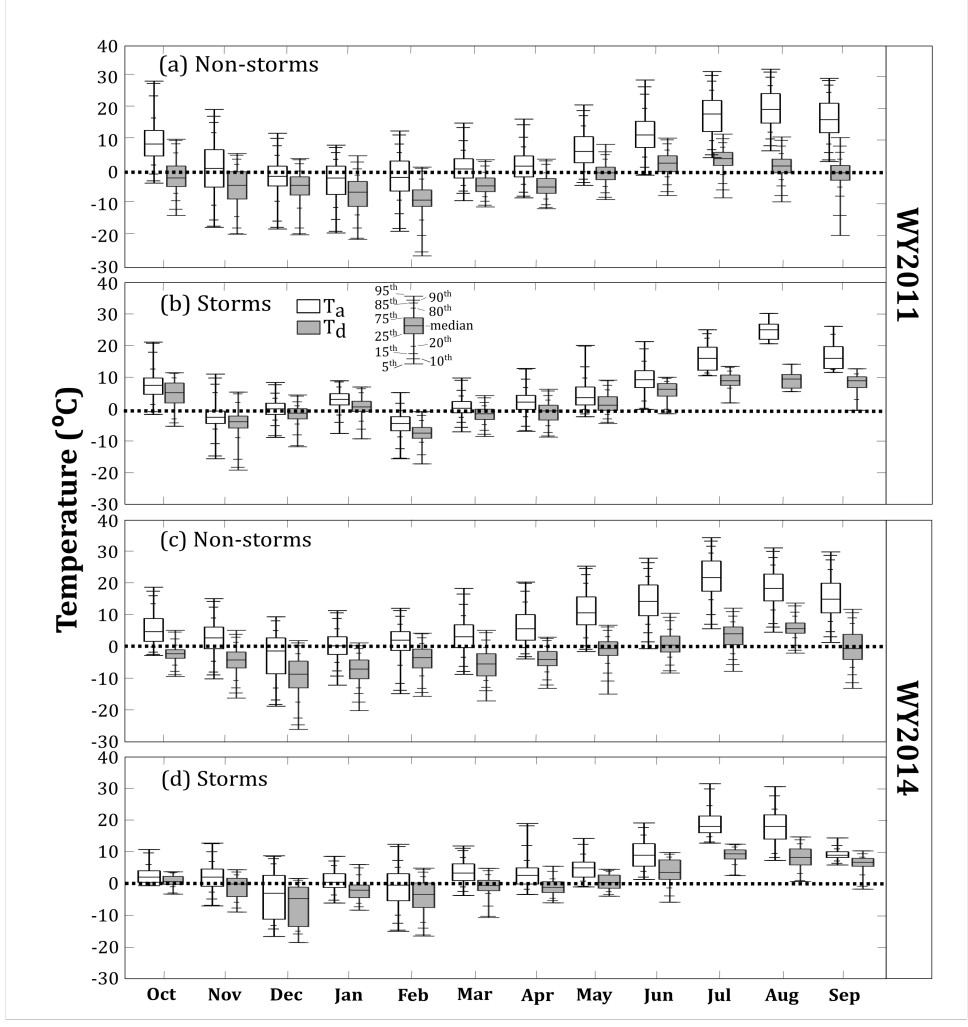

**Figure 2: Monthly average $T_a$ (white) and $T_d$ (grey) during non-storm (a and c) and storm (b and d) periods for WY2011 and WY2014 based on data from all stations. The legend in (b) applies to all panels; boxes depict the interquartile range and whiskers indicate the 5th and 95th percentiles of data.**





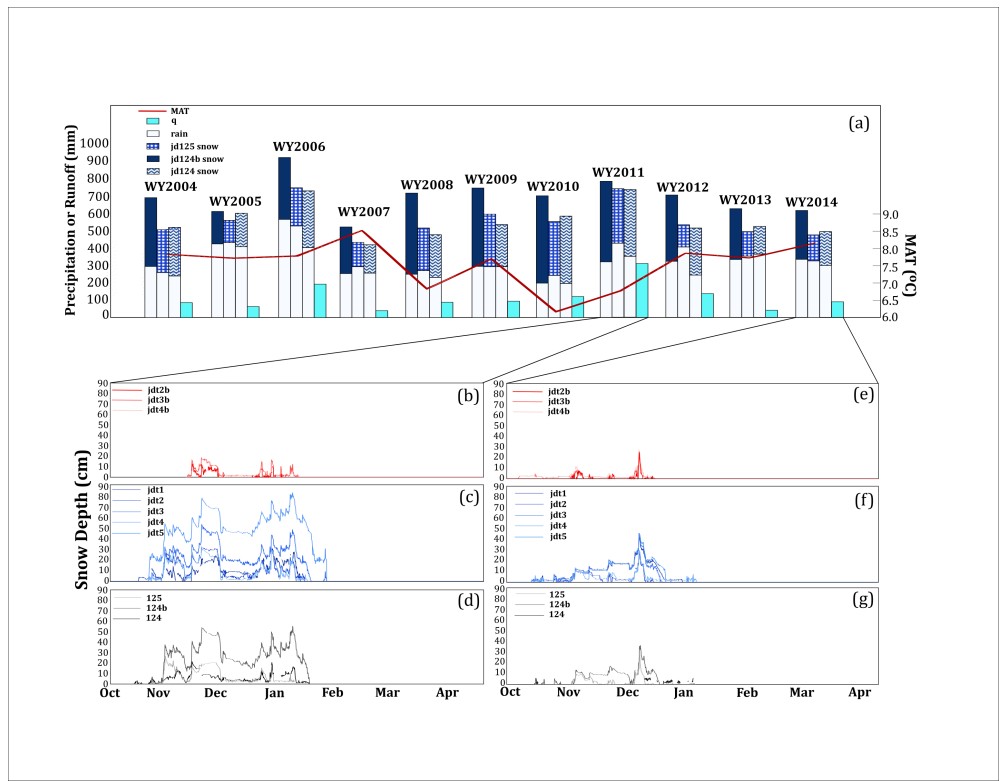

**Figure 3: (a) Cumulative annual precipitation from sites 125, 124b, and 125 and cumulative runoff (q) as measured at the JD outlet and mean annual temperature (MAT) based on stations 124 and 125 which are collecting data over the entire 11 WY. That the three precipitation sites are similar in WYs 2005 & 2011 suggests warmer, less windy conditions. Streamflow (q) is specific discharge, or total volumetric stream flow (Q) normalized by catchment area. (b–g) Snow depth in the JD for WY2011 (b-d) and WY2014 (e-g). Panels (b) and (e) show snow depth from sites on the south-facing slopes in red colors, (c) and (f) show snow depth on the north-facing slopes in blue colors, and (d) and (g) show snow depth from 125, 124b, and 124 in grayscale. Each legend shows sites ordered from lowest to highest elevations with brighter tones at lower elevations. Snow depths increase inconsistently with increasing elevations due to wind scour from exposed sites and accumulation in sheltered areas. Gaps due to instrument failure are seen as breaks in the continuous time series line in some subplots.**




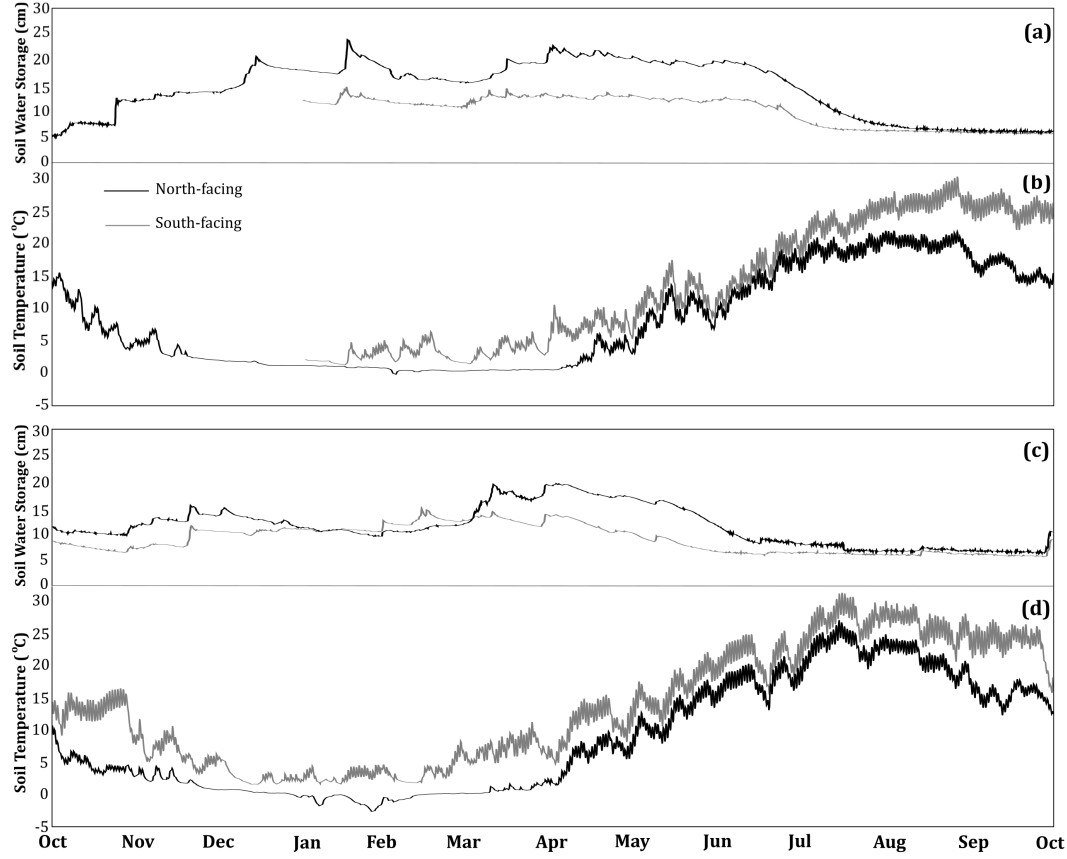

**Figure 4. Soil water storage (panels (a) and (c)) and ground temperatures at 20cm (panels (b) and (d)) on both north- and south-facing slopes, up to depths of 100 and 50 cm, respectively, in WY 2011 (a-b) and 2014 (c-d). The legend in (b) applies to all panels. Water storage and temperatures for north-facing slopes were calculated based on jdt1, jdt2, jdt3, and jdt4, and for south-facing slopes, these values were calculated based on jdt2b, jdt3b, and jdt4b. During October - January of WY2011 (panel a), no soil temperature or moisture data on the south-facing slopes were collected due to the soil moisture sensors not yet being installed.**



| Station | Elevation (m) | Aspect | Start Date | Duration (WY) | $T_a$ | | | $T_d$ | | | $w_s$ | | | $w_d$ | | | $ppt_a$ | $z_s$ | $T_g$ | $\theta$ | $S_i$ | $Q$ |
|---|---|---|---|---|---|---|---|---|---|---|---|---|---|---|---|---|---|---|---|---|---|---|
| | | | | | $\overline{WY}$ | $\overline{S}$ | $\overline{NS}$ | $\overline{WY}$ | $\overline{S}$ | $\overline{NS}$ | $\overline{WY}$ | $\overline{S}$ | $\overline{NS}$ | $\overline{WY}$ | $\overline{S}$ | $\overline{NS}$ | $\overline{WY}$ | $\overline{WY}$ | $\overline{WY}$ | $\overline{WY}$ | | $\overline{WY}$ |
| *125b* | 1496 | NE | 2003-01-10 | 11 | | | | | | | | | | | | | | | | | | 114 |
| *125* | 1508 | SE | 2003-01-10 | 11 | 8.1 | 3.7 | 9.1 | -1.6 | -0.2 | -1.9 | 1.8 | 2.2 | 1.7 | 195 | 213 | 191 | 564 | 21 | | | 14.9 | |
| *jdt1* | 1552 | N | 2005-05-11 | 9 | 8.6 | 3.2 | 9.9 | -2 | -0.7 | -2.3 | | | | | | | | 41 | 8.8 | 0.12 | | |
| *jdt2b* | 1611 | S | 2011-03-04 | 4 | 9.1 | 4.0 | 10.3 | -1.8 | -0.3 | -2.2 | 2.8 | 3.7 | 2.6 | 211 | 212 | 210 | | 5 | 12.3 | 0.23 | | |
| *jdt2* | 1613 | N | 2005-11-05 | 9 | 8.4 | 2.9 | 9.7 | -2.6 | -1.3 | -2.9 | | | | | | | | 31 | 7.2 | 0.12 | | |
| *jdt3* | 1655 | N | 2005-09-21 | 9 | 8.2 | 2.6 | 9.5 | -2.6 | -1.3 | -3.0 | 2.7 | 2.9 | 2.7 | 206 | 237 | 198 | | 71 | 7.4 | 0.14 | | |
| *jdt3b* | 1659 | S | 2010-12-13 | 4 | 8.4 | 3.3 | 9.6 | -2.0 | -0.5 | -2.5 | 3.1 | 3.8 | 2.9 | 208 | 214 | 207 | | 12 | 12.7 | 0.15 | | |
| *jdt4b* | 1704 | S | 2011-03-04 | 4 | 8.8 | 3.4 | 10.1 | -2.2 | -0.6 | -2.6 | 2.9 | 4.0 | 2.6 | 225 | 228 | 224 | | 14 | 12.5 | 0.15 | | |
| *jdt4* | 1706 | N | 2005-11-02 | 9 | 8.0 | 2.2 | 9.5 | -2.6 | -1.3 | -3.0 | | | | | | | | 113 | 6.6 | 0.12 | | |
| *jdt5* | 1757 | N | 2005-11-02 | 9 | 7.4 | 1.9 | 8.7 | -2.7 | -1.5 | -3.0 | | | | | | | | 38 | | | | |
| *124b* | 1778 | SE | 2006-11-11 | 8 | 6.9 | 1.8 | 8.2 | -2.1 | -1.1 | -2.4 | 1.8 | 2.4 | 1.7 | 217 | 232 | 213 | 700 | 70 | 8.2 | 0.21 | 16.9 | |
| *124* | 1804 | NE | 2003-10-1 | 11 | 7.0 | 1.5 | 8.4 | -2.6 | -1.6 | -2.8 | 4.5 | 6.7 | 3.9 | 218 | 240 | 213 | 563 | 20 | | | 16.7 | |
| **Average** | | | | | 8.1 | 2.8 | 9.4 | -2.3 | -1.0 | -2.6 | 2.8 | 3.7 | 2.6 | 211 | 225 | 208 | 609 | 40 | 9.5 | 0.2 | 16.2 | 114 |

Table 1. Stations within Johnston Draw watershed, their elevations, and available parameters at each station. Although data presented in this dataset are limited to the end of WY2014 (2014-09-30), data from all stations continues further than WY2014 as all stations are currently maintained. Full station names used by the USDA ARS and recorded in the published data set include the prefix "rc.tg.dc.jd-" before each abbreviated station name recorded in the 1st column of this table, however for simplicity, these abbreviated station names are used throughout this manuscript. Full naming convention is provided in the Naming Convention file with the published dataset. Because $T_d$ was calculated based on RH (see text for details), RH is not summarized here, but is available at all stations except 125b (streamflow station). Start date is presented as YYYY-MM-DD. $T_a$ – air temperature (°C), $T_d$ – dewpoint temperature (°C), $w_s$, $w_d$ – wind speed (ms$^{-1}$), wind direction (0 - 359°, 0 = north; 180 = south), $S_i$ – incoming solar radiation (MJ m$^{-2}$ day$^{-1}$), $ppt_a$ – wind-corrected precipitation (mm), $z_s$ – snow depth (mm), $T_g$ – soil temperature at 20 cm depth (°C), $\theta$ – soil water or moisture at 20 cm depth (m$^3$ m$^{-3}$), Q – streamflow (mm), $\overline{WY}$ = mean WY value, $\overline{S}$ = mean storm value and $\overline{NS}$ =mean non-storm value. $\overline{WY}$ for $T_g$ and $\theta$ at 124b were calculated based on the two subsurface measurement locations at this site (see text for details). Note that $S_i$ and Q units have been converted from the database to the units reported here for comparison across parameters and sites. Snow depth is averaged over WY 2011-2014 for all sites to facilitate meaningful comparisons.