# Peer review of "Earth Syst. Sci. Data Discuss., https://doi.org/10.5194/essd-2017-112 Manuscript under review for journal Earth Syst. Sci. Data"

_Earth System Science Data, 2017_

## Referee Comment (RC1) · U. Strasser (Referee) · 10 Dec 2017

This paper represents a very valuable data contribution to hydrometeorological assessemts of the rain-to-snow transition zone at the catchment scale. The presented data is unique, useful and complete in the sense of the ESSD review criteria. I recommend publication in ESSD after some minor improvements.

- p. 2, abstract, line 6: please provide a clickable doi (as in line 7 on p. 9)

- p. 2, abstract, line 10: better "air" temperature

- p. 2, abstract, line 12: better specifiy for which type of models the provided datasets are useful (e.g., "hydrological and boundary layer flux" models?)

- p. 2, introduction, line 26: take out "…, and varies in both time and space." (is repeated in next sentence)

- p. 2, introduction, line 30: better " important to be studied…"?

- p. 2, introduction, line 31: better "… frequently experience winter temperatures…"?

- p. 3, introduction, line 9: for the final version of the manuscript the keyword search should be updated

- p. 3, introduction, lines 8 to 21: Your web search may show how much the terms You searched for are used by the scientific community, but not how limited the transition zone datasets really are. No wonder that Your search only provides one result from the European Alps, the "transition zone" term being less prominent there than in the U.S. For sure there are several research catchments in the Alps with basin-wide measurements and stream discharge, they are published only with other than Your search terms. Nevertheless, this does not reduce the significance of Your data, and Your work! Maybe You better move this aspect to the end of the paper, to rise awareness for the importance of the transition zone catchments for future hydrometeorological research?

- p.3, introduction, line 27: better "air" temperature

- p.3, introduction, line 29: what do You mean with "… possibly representing what can be expected as regional climate warming advances."? Your data has been observed, hence it represents historical/current conditions, and nothing that "can be expected". Better specify.

- p. 13, Fig. 1: better take out the color scheme for elevation in the map; the colors are unusual for the purpose, ambiguous, and contour lines are hardly visible
- p. 17: Table 1: insert the missing "0" in the Start Date of station 124. Better provide full station names in the table as used by USDA ARS. Correct order of meteorological variables in the caption, according to the table. Provide correct assignment of abbreviation and explanation (for wind speed) in the caption.

- p. 4-5, site description, second paragraph: this paragraph probably better fits to the beginning of section 3 (Data Description)

- p. 14, Fig. 2: better take out legend from (b) and place it near the panels

- p. 6, radiation, line 10: indicate which type of model You mean

- p. 6, radiation, line 9-15: You mention the importance of longwave radiation for energy balance applications, but You do not consider it to be measured in JD in the future. Why?

- p. 6, precipitation, lines 25-30: You explain the different methods used for wind correction of precipitation. Are the raw rain gauge recording still available, too? This is a general question that I recommend to be considered: since the raw (logger) recordings are available, You could include a short note on this in the beginning of the data section of the paper

- p. 15, Fig. 3: replace "&" with "and". Consider to replace the tone scheme to indicate the station elevation with colors. The aspect to which the three panels for Snow Depth belong to can be better indicated with "south-facing" and "north-facing", instead of the colors. It seems that blowing snow is a major issue in JD, and should be investigated with its implications on snowmelt patterns and runoff generation. I recommend to add respective considerations in the paper

- p. 7, Snow Depths, line 16: better use singular: "Snow Depth"

- p. 8, Snow Depths, lines 3-9: methods to convert snow depth to swe require observations or estimates of snow density. It would be helpful to indicate this for both the LIDAR techniques, and snow modelling

- p. 16, Fig. 4: better take out legend from (b) and place it near the panels. Better indicate the WY directly in the panels

- p. 9, Data Availability, lines 7-13: You should mention here that original text files for (i) soil moisture, temperature and snow depth, (ii) precipitation and (iii) weather data are available as well

- p. 9, Conclusions, line 15: You claim this, but You cannot know. Better add something like "to the knowledge of the authors"

- p. 9, Conclusions, line 18: add "soil" to temperature

- p. 9, Conclusions, line 21: add what type of models You mean with "a variety of models"

- p. 9, Conclusions, line 22: explain what You mean with "basin-scale interactions and responses"

- p. 9, Conclusions, lines 23-27: these final sentences better fit into the introduction

- p. 9, Conclusions, line 27: eleven years of data not yet allow for the assessment of climate change impacts, better say "changes in meteorological conditions. . ."

Thank You for considering me as a reviewer, and good luck!

---

## Referee Comment (RC2) · J. Dozier (Referee) · 2 Jan 2018

The manuscript makes an excellent case for the possible uniqueness of this dataset.

Page 2, Line 31; and Page 5, Line 18: "near freezing": I think you mean "near melting." $0°C$ is the melting temperature of ice, whereas the freezing temperature of water in the atmosphere can be as low as $-40°C$.

Page 6, Lines 13-15: For clear skies, it would be useful to calibrate Prata's [1996] equation against the longwave data from elsewhere in the RCEW. Generally, we don't measure precipitable water, instead we (and Prata) infer it from surface vapor pressure. I'm not sure of the elevations of the data that Prata used, but my own experience in comparing the data at CUES (also in this issue of ESSD) is that Prata's equation systematically over-estimates longwave radiation at the CUES elevation. The form of the equation is fine, but the coefficients should be adjusted where nearby data are available.

Page 6, Lines 25-31: Can you expand the text here slightly to explain how the various estimates and corrections work? In the current form, I have to read Marks et al. [1999, 2013], Hanson et al. [2004], Hamon [1973], and WMO [2008] to have even the faintest idea of what's involved. I'd be happy to go to the original references for the details, but a few clauses here would help. You provide a goo example, on the next page (Page 7, Lines 19-21). I don't have to read Ryan et al. [2008] to get a basic understanding of how the data are filtered.

Page 8, Line 4: Jeff Deems wants us to use "lidar" instead of "LiDAR." Although I don't have a strong opinion, I do point out to students that we use "radar" not "RaDAR."

Page 9, Line 8: I've examined the website https://doi.org/10.15482/usda.adc/1402076. In the polygonal definition of the coverage, it's not necessary to express latitude and longitude to 11 decimal places, about 1 $\mu$m on Earth's surface. On a positive note, the descriptions of the data are complete enough to unambiguously convert the CSV files to software for analysis.

---

## Referee Comment (RC3) · Anonymous Referee #3 · 2 Jan 2018

This paper presents a detailed hydro-meteorological dataset from a small catchment in the rain-to-snow transition zone in southwestern Idaho, USA. This dataset covers a wide range of altitude and aspect across the rain-to-snow transition zone. The paper is well written and the data are easy to access in a convenient format on the USDA data website with a complete description of the metadata. Therefore, I recommend the publication of this paper in ESSD subject to minor revisions outlined below.

[Figure]

Specific comments

P 2 L 26: the extension of the rain-to-snow transition zone in the Northwestern US in terms of km2 does not mean a lot for the reader who is not familiar with this region of the world. The authors could for example give the relative importance of the rain-to-snow transition for the mountains of the Northwestern US.

P 3 L 7-L21: the results of the meta-analysis is interesting but I am wondering if the keywords used by the authors are sufficient to get a clear overview of the dataset available from sites or catchments lying in the rain-to-snow transition zone across the world. Among the 5 sites listed, 3 of them are located in the northwestern US (with two of them in southwestern Idaho). Does it mean that the terms "rain-to-snow transition zone" is mainly used in the US? For example, as mentioned in the paper, the Col de Porte experimental site in the French Alps is typically lying in this zone but wasn't discovered when searching the keyword.

P3 L 14: the site in Davos is the Weissfluhjoch test site managed by SLF. This site is located at 2540 m in the Swiss Alps. Can it be reasonably classified in the rain-to-snow transition zone ?

P3 L 20-21: Note that Col de Porte and Weissfluhjoch are mainly reference sites for snow observations and one of their main objective is to provide atmospheric forcing and detailed evaluation data for snowpack models. This objective is different from this dataset that provides distributed hydro-meteorological data from a small catchment in the rain-to-snow transition zone.

P 3 L 27: add "and melting" after "snow accumulation"

P3 L 28-30: this dataset concerns the present climate and it is hard to say that it is possibly representing the future evolution. I recommend the authors to remove this sentence. Also, the dataset only covers 11 years which is not along enough from a climate perspective.

P 4 L 11: what are the typical slope angles found on the south-facing and north-facing slopes?

P 4 L 23: "z_s" is not a classical symbol for snow depth. Consider using the symbol from the international classification for seasonal snow on the ground (Table 2.1 in Fierz et al. 2009)

P 5 L 9: are the data from stations 144 and 145 available as well? At P 6 L 13, the author mention a dataset in preparation by Marks et al. Is it the same dataset?

P 5 L 10-11: is there a flag in the dataset that mentions the time periods when gaps have been filled?

P 5 L 13-14: you could refer here to Fig. 3 that shows a nice overview of averaged precipitation and temperature during the 11-WY time period

P 6 L 7: Could the authors include a brief comment on the influence of the surrounding topography on incoming SW? For example, are they shadows from the surrounding topography that modifies incoming SW measured at the stations in early morning or late afternoon? Overall, if available, it would be interesting to know the topographic mask of each station with local horizon angles.

P 6 L 17: it would be interesting to know at which height above the ground are typically measured wind speed and if snow depth is measured at all stations measuring wind speed. This information is useful to know at which height above the snow surface wind speed is measured in wintertime.

P 6 L 18: can the author add a comment about the representativeness of wind speed measurement? Does the surrounding vegetation influence wind speed measurement at some stations?

P 6 L 25: are the raw precipitation data included in the dataset ? It would be interesting to have them if data users want to apply their own methods of correction following for example the recent SPICE project.

P 8 L 2: based on Fig. 3, it appears that wind-induced snow transport strongly affects snow depth evolution at some stations. Could the authors comment more on the influence of wind-induced snow transport on the seasonal evolution of snow depth at this site? What are the stations that are typically exposed to wind-induced erosion and accumulation?

P 8 L 4: can the author comment more on the method use to convert snow depth to SWE? Do they mean using the bulk snowpack density simulated by a snowpack scheme to convert measured snow depth into estimated SWE?

P 8 L 31: At P4 L3 the authors mention that the size of the catchment is 1.79 km2 which differs from 181 ha. The differences are small but what is the actual area of the catchment?

P 9 L 15-16: it would interesting to mention in the conclusion that data are still collected at this experimental catchments and to precise whether the dataset will be updated on a regular basis to include the more recent years.

Figure 1: terrain contour lines are hard to read.

Figure 3 is very interesting and nicely shows the different temperature and precipitation conditions at for this catchment. However, the graphics showing snow depth evolution are hard to read. Maybe make two separate figures. The snow depth times series goes from October to late April whereas all the snow is generally gone in February. This graphics would be easier to read if they showed snow depth evolution from October to late February.

References

Fierz, C. R. L. A., Armstrong, R. L., Durand, Y., Etchevers, P., Greene, E., McClung, D. M., ... & Sokratov, S. A. (2009). The international classification for seasonal snow on the ground (Vol. 25). Paris: UNESCO/IHP.

---

## Author Comment (AC1) · 8 Feb 2018

Note: a formatted response to reviewers' comments is available in the supplemental file

Reviewer #1 (Strasser): This paper represents a very valuable data contribution to hydrometeorological assessemts of the rain-to-snow transition zone at the catchment scale. The presented data is unique, useful and complete in the sense of the ESSD

review criteria. I recommend publication in ESSD after some minor improvements. Thank you. We respond to the specific recommendations for improvements below. - p. 2, abstract, line 6: please provide a clickable doi (as in line 7 on p. 9) Done. - p. 2, abstract, line 10: better "air" temperature This is actually intended to refer to soil rather than air temperatures, and we have added another "soil" modifier to the sentence to clarify. - p. 2, abstract, line 12: better specifiy for which type of models the provided datasets are useful (e.g., "hydrological and boundary layer flux" models?) We have added "hydrological" before models here. - p. 2, introduction, line 26: take out "..., and varies in both time and space." (is repeated in next sentence) To clarify the key idea here, we combined the key points into one sentence, which now reads, "This broad characterization is not stationary in space or time, however, and its extent varies with climate conditions, latitude, and distance from the ocean." - p. 2, introduction, line 30: better " important to be studied. . ."?
- We think that the current phrasing is clearer, expressing the same idea more concisely. - p. 2, introduction, line 31: better ". . . frequently experience winter temperatures. . ."? Agreed. We have added this adverb. - p. 3, introduction, line 9: for the final version of the manuscript the keyword search should be updated We repeated this analysis on 2/5/18. Although there are 12 additional returns since the original search, there are no additional published datasets; we updated the text to reflect this information: "We did this by searching the key words: "rain-snow transition data" OR "rain snow zone data" in Web of Science (search date: 2/5/2018). Out of the 91 returns, only 5 publications (5.4%) had published freely available hydrometeorological data in the rain-to-snow transition zone." - p. 3, introduction, lines 8 to 21: Your web search may show how much the terms You searched for are used by the scientific community, but not how limited the transition zone datasets really are. No wonder that Your search only provides one result from the European Alps, the "transition zone" term being less prominent there than in the U.S. For sure there are several research catchments in the Alps with basin-wide measurements and stream discharge, they are published only with other than Your search terms. Nevertheless, this does not reduce the significance of Your data, and Your work! Maybe You better

move this aspect to the end of the paper, to rise awareness for the importance of the transition zone catchments for future hydrometeorological research? We agree with the reviewer that our search will not reveal other studies that occur in this zone that have not used these terms in a way that Web of Science will discover them, as we state on lines 15-16. We do not think that this result necessarily encompasses all global work in this zone, but data that is difficult to discover using Web of Science may be less used by modelers and other researchers. We highlight the term and this meta-analysis here to (hopefully) encourage wider adoption to facilitate data discovery and comparisons. - p.3, introduction, line 27: better "air" temperature As above, this correctly refers to soil temperature, so we have not changed the text. - p.3, introduction, line 29: what do You mean with ". . . possibly representing what can be expected as regional climate warming advances."? Your data has been observed, hence it represents historical/current conditions, and nothing that "can be expected". Better specify. We intended here to point out that although the record at this location is only 11 years long, these years are relatively warm based on much longer-term (30+ year) records collected nearby. To better guide the reader, we have added two citations that address the Reynolds Creek Experimental Watershed-scale, long-term record and trend analysis. - p. 13, Fig. 1: better take out the color scheme for elevation in the map; the colors are unusual for the purpose, ambiguous, and contour lines are hardly visible We have used a typical color ramp for the elevation range, and aim here to provide a quick visualization of the available elevation information and context as well as the observation locations. We note that the DEM shapefiles are available in the published dataset if other users want to use a different color scheme. - p. 17: Table 1: insert the missing "0" in the Start Date of station 124. Better provide full station names in the table as used by USDA ARS. Correct order of meteorological variables in the caption, according to the table. Provide correct assignment of abbreviation and explanation (for wind speed) in the caption. Thank you. We have corrected the start date and order/splitting of variables in the caption. Although it might be easier for some readers if we included the full ARS station name in the table, these names are so long that the full table would not fit on one page.

We decided that it would be better to include the full station name in the caption and to fit all the information on one page to serve as a reference for most readers. - p. 4-5, site description, second paragraph: this paragraph probably better fits to the beginning of section 3 (Data Description) We see your point and moved this paragraph to a new section 3.1, "Instrumentation Overview" and updated all subsequent section numbers. - p. 14, Fig. 2: better take out legend from (b) and place it near the panels We included it within panel (b) to maximize the size of the panels for each figure. To move the legend outside all panels would require shrinking the overall figure size, which we think is not justified. - p. 6, radiation, line 10: indicate which type of model You mean We added the phrase, "when assessing snowmelt timing, peak snow water equivalent, and snow surface temperatures" to clarify this point. - p. 6, radiation, line 9-15: You mention the importance of longwave radiation for energy balance applications, but You do not consider it to be measured in JD in the future. Why? Good point. We have added the phrase, "and may be added to JD in the future" to the manuscript on line 17. - p. 6, precipitation, lines 25-30: You explain the different methods used for wind correction of precipitation. Are the raw rain gauge recording still available, too? This is a general question that I recommend to be considered: since the raw (logger) recordings are available, You could include a short note on this in the beginning of the data section of the paper This information can be made available to those who contact the authors or the ARS directly, but because we have spent many hours quality checking the data, and have carefully considered the wind-correction algorithms we applied, we believe that we are providing the best-quality data available. - p. 15, Fig. 3: replace "&" with "and". Consider to replace the tone scheme to indicate the station elevation with colors. The aspect to which the three panels for Snow Depth belong to can be better indicated with "south-facing" and "north-facing", instead of the colors. It seems that blowing snow is a major issue in JD, and should be investigated with its implications on snowmelt patterns and runoff generation. I recommend to add respective considerations in the paper We replaced the ampersand with "and". We used the shades to visually indicate that higher elevations (lighter shades) tended to have deeper snowpacks, and that the

differences between WY2011 and WY2014 were nearly as large as the differences among sites on the elevation gradient. Because the north-facing snow depths are so large in WY2011 compared to the south-facing sites in either year or across all aspects in WY 2014, changing the colors does little. Further, changing the y-axis limits across aspect and year would minimize the very effect that we are trying to show. However, we have truncated the x-axes in all subpanels to help expand the visible area. For those who might be interested in seeing other comparisons, we hope that they will use the dataset! We also agree that blowing snow may be important at certain times within parts of JD, and we have added a sentence alluding to possible future work that may address this issue (p. 8, lines 13-15): "Wind redistribution of blowing snow is known to affect the nearby Reynolds Mountain East (RME) catchment (Winstral et al. 2012) and may also be important at times in JD; this data set facilitates further exploration of wind effects at the rain-snow transition on melt patterns and runoff generation." - p. 7, Snow Depths, line 16: better use singular: "Snow Depth" By using the plural, we are trying to convey that depth was measured at many locations, not just one. - p. 8, Snow Depths, lines 3-9: methods to convert snow depth to swe require observations or estimates of snow density. It would be helpful to indicate this for both the LIDAR techniques, and snow modelling We have added "or snow density" to page 8, line 16. - p. 16, Fig. 4: better take out legend from (b) and place it near the panels. Better indicate the WY directly in the panels As above, we prefer to include the legend within the panel to make the figure as large as possible. We have added the WY labels to the panels as suggested. - p. 9, Data Availability, lines 7-13: You should mention here that original text files for (i) soil moisture, temperature and snow depth, (ii) precipitation and (iii) weather data are available as well We are unsure what else would be helpful here since the statement already says, "All data presented in the paper are available" and each of the requested items is explicitly listed including how many files are available for each in lines 22-24. - p. 9, Conclusions, line 15: You claim this, but You cannot know. Better add something like "to the knowledge of the authors" We have added, "To our knowledge," at the start of this sentence. Based on our search of the literature, we have

not found another published dataset that includes all of the elements in this dataset for this record length. - p. 9, Conclusions, line 18: add "soil" to temperature We switched the order of soil temperature and moisture to attempt to alleviate any confusion. - p. 9, Conclusions, line 21: add what type of models You mean with "a variety of models" We added the adjective "hydrometeorological" to modify models. - p. 9, Conclusions, line 22: explain what You mean with "basin-scale interactions and responses" We were thinking about interactions such as increased temperature and radiation inputs leading to heterogeneous snowpack accumulation and melt rates, and associated shifts in soil temperature/moisture and runoff. These interactions and responses can be assessed at the integrated scale of the basin with this dataset, but also at a finer spatial resolution as well. - p. 9, Conclusions, lines 23-27: these final sentences better fit into the introduction
 Although we agree that they could also be mentioned in the introduction, we hope to wrap up the paper with a call to possible future uses of this exciting data set. - p. 9, Conclusions, line 27: eleven years of data not yet allow for the assessment of climate change impacts, better say "changes in meteorological conditions. . ." Good point. We were thinking (very far) ahead here; we have made this suggested change to: "variability in meteorological conditions..." Thank You for considering me as a reviewer, and good luck! Thank you for your helpful suggestions and feedback.

Reviewer #2 - Dozier

The manuscript makes an excellent case for the possible uniqueness of this dataset. Thank you. We hope it will be helpful to the community.

Page 2, Line 31; and Page 5, Line 18: "near freezing": I think you mean "near melting." 0 ◦C is the melting temperature of ice, whereas the freezing temperature of water in the atmosphere can be as low as -40◦C.

Yes, we did intend to refer to the phase transition near 0°C, and have simply revised these two sections to refer to the temperature instead of stating the freeze/thaw processes that may or may not be occurring at exactly that temperature.

Page 6, Lines 13-15: For clear skies, it would be useful to calibrate Prata's [1996] equation against the longwave data from elsewhere in the RCEW. Generally, we don't measure precipitable water, instead we (and Prata) infer it from surface vapor pressure. I'm not sure of the elevations of the data that Prata used, but my own experience in comparing the data at CUES (also in this issue of ESSD) is that Prata's equation systematically over-estimates longwave radiation at the CUES elevation. The form of the equation is fine, but the coefficients should be adjusted where nearby data are available.

Thank you for this new information. We have added the following sentence to help point readers in the correct direction, "New work (Bair et al., this issue) suggests that some of these calculations may be sensitive to elevation and should be calibrated against nearby measurements, if possible."

Page 6, Lines 25-31: Can you expand the text here slightly to explain how the various estimates and corrections work? In the current form, I have to read Marks et al. [1999, 2013], Hanson et al. [2004], Hamon [1973], and WMO [2008] to have even the faintest idea of what's involved. I'd be happy to go to the original references for the details, but a few clauses here would help. You provide a goo example, on the next page (Page 7, Lines 19-21). I don't have to read Ryan et al. [2008] to get a basic understanding of how the data are filtered.

Thank you for this suggestion to improve ease of access for readers to the corrections that were made. We have revised the section to read, "The dataset includes wind-corrected (ppta) precipitation measurements for three sites in JD (125, 124, 124b) and the percentage of precipitation that is in the form of rain, snow, or a mixture of rain and snow. The latter were calculated using the humidity-based methods developed by Marks et al. [1999, 2013], where Td values below -0.5°C are considered all snow, above +0.5°C are considered all rain, with a linear ratio of mixed rain/snow between these thresholds. The precipitation data for stations 125 and 124 were wind-corrected using a dual-gage correction method developed at RCEW [Hamon et al. 1973; Hanson

et al., 2004], whereby wind-corrected precipitation is an empirical function of the ratio between unshielded and shielded gage catch. Because the 124b site has only a single gage, the dual gage correction methods cannot be applied to this site. Instead the shielded data for 124b were wind-corrected using WMO [2008] methods, where the corrected precipitation mass is a function of the wind speed and precipitation phase."

Page 8, Line 4: Jeff Deems wants us to use "lidar" instead of "LiDAR." Although I don't have a strong opinion, I do point out to students that we use "radar" not "RaDAR."

The preferred capitalization appears to be evolving; we have adopted your and Deems' recommendation here.

Page 9, Line 8: I've examined the website https://doi.org/10.15482/usda.adc/1402076. In the polygonal definition of the coverage, it's not necessary to express latitude and longitude to 11 decimal places, about 1 $\mu$m on Earth's surface. On a positive note, the descriptions of the data are complete enough to unambiguously convert the CSV files to software for analysis. Thank you. The stated boundaries are stated using the default precision for the data library coverage area; we agree that we do not have micron-scale data. The library is evaluating whether to change this default, and for now we simply note that the GIS layers that are available online are correct, and that users may need to use discretion in believing the reported precision.

Reviewer #3 (anon.): This paper presents a detailed hydro-meteorological dataset from a small catchment in the rain-to-snow transition zone in southwestern Idaho, USA. This dataset covers a wide range of altitude and aspect across the rain-to-snow transition zone. The paper is well written and the data are easy to access in a convenient format on the USDA data website with a complete description of the metadata. Therefore, I recommend the publication of this paper in ESSD subject to minor revisions outlined below. Thank you for this summary and recommendation.

Specific comments P 2 L 26: the extension of the rain-to-snow transition zone in the Northwestern US in terms of km2 does not mean a lot for the reader who is not familiar

with this region of the world. The authors could for example give the relative importance of the rain-to-snow transition for the mountains of the Northwestern US. Good point. We have revised this section to provide more context: "covers ∼1% of total land area in the region".

P 3 L 7-L21: the results of the meta-analysis is interesting but I am wondering if the keywords used by the authors are sufficient to get a clear overview of the dataset available from sites or catchments lying in the rain-to-snow transition zone across the world. Among the 5 sites listed, 3 of them are located in the northwestern US (with two of them in southwestern Idaho). Does it mean that the terms "rain-to-snow transition zone" is mainly used in the US? For example, as mentioned in the paper, the Col de Porte experimental site in the French Alps is typically lying in this zone but wasn't discovered when searching the keyword. Indeed this may be the case. The Col de Porte dataset is obviously an important resource as well, and we explicitly point it out in this overview of the meta-analysis because of its importance. We looked to see whether that paper incorporated a set of keywords that we could use to expand our search terms, but it did not. Searching for the terms "mid-altitude" or "mid-altitude mountain" (used in the title) and "data" resulted almost exclusively in ecological studies, so we did not add these results to the revised manuscript. We would be happy to add more keywords to our own paper and this meta-analysis if the reviewer has additional suggestions.

P3 L 14: the site in Davos is the Weissfluhjoch test site managed by SLF. This site is located at 2540 m in the Swiss Alps. Can it be reasonably classified in the rain-to-snow transition zone ? The mixed precipitation zone likely varies in different regions, but this elevation falls outside of the mixed precipitation zone identified by Tennant et al. (2015) in the US Intermountain West of ∼1500-2250m based on analysis of precipitation phase partitioning. More detailed analysis of the precipitation at that site would be warranted to evaluate the proportions of rain and snow, but that is outside the scope of this data paper from a different location.

P3 L 20-21: Note that Col de Porte and Weissfluhjoch are mainly reference sites for snow observations and one of their main objective is to provide atmospheric forcing and detailed evaluation data for snowpack models. This objective is different from this dataset that provides distributed hydro-meteorological data from a small catchment in the rain-to-snow transition zone. We fully agree with the reviewer here. Each of these sites has been designed for different purposes, and the JD site does not include all the information described in the Col de Porte data paper. We attempt to point out the differences between the datasets here simply to highlight the unique aspects of the JD.

P 3 L 27: add "and melting" after "snow accumulation" Done.

P3 L 28-30: this dataset concerns the present climate and it is hard to say that it is possibly representing the future evolution. I recommend the authors to remove this sentence. Also, the dataset only covers 11 years which is not along enough from a climate perspective. As noted above in response to a previous reviewer's comment, we added citations for two other papers that have demonstrated longer-term climatic trends (Nayak et al. 2010, and Kormos et al., this issue) at the larger Reynolds Creek Experimental Watershed/CZO, of which JD is a part. Those trends indicate that the shorter JD record encompasses a warmer set of years than used to be common.

P 4 L 11: what are the typical slope angles found on the south-facing and north-facing slopes?

We have addressed this point by adding, "North-facing slopes are slightly steeper with an average slope of 16.8° whereas the average south-facing slopes are 13.9° (Patton, 2016)."

P 4 L 23: "z_s" is not a classical symbol for snow depth. Consider using the symbol from the international classification for seasonal snow on the ground (Table 2.1 in Fierz et al. 2009)

Although we recognize the value of this new international classification scheme, we
established this dataset well before the scheme was adopted, and all Agricultural Research Service (ARS) databases for the experimental watershed refer to snow depth as z_s. We have edited the text to note the equivalence between the established ARS symbols and the international standard, "…equivalent to HS in the International Seasonal Snow Classification established by Fierz et al. 2009…".

P 5 L 9: are the data from stations 144 and 145 available as well? At P 6 L 13, the author mention a dataset in preparation by Marks et al. Is it the same dataset?

The data from stations 144 and 145 are located outside of JD, and are thus not included in this dataset. They were included in the dataset published by Kormos et al. in this issue and are available through the ARS. The dataset mentioned on p. 6 refers to another station (#176) that is also incorporated into the Kormos et al. dataset for the larger Reynolds Creek Experimental Watershed.

P 5 L 10-11: is there a flag in the dataset that mentions the time periods when gaps have been filled?

Unfortunately, we did not flag the datasets as they were processed to distinguish between corrections introduced to the QC process, gap-filling, or wind corrections for the precipitation. Instead of providing a generic flag to show change, we note in the revised manuscript that only up to ∼1% of the dataset differs from the raw dataset, "Because additional sites were added during the period of record, sometimes gaps were filled by different neighbouring sites during different periods; during the periods reported as active for each station in Table 1, up to ∼1% of records were gap-filled or corrected."

P 5 L 13-14: you could refer here to Fig. 3 that shows a nice overview of averaged precipitation and temperature during the 11-WY time period We have added this reference: "(see Figure 3 for range of conditions)."

P 6 L 7: Could the authors include a brief comment on the influence of the surrounding topography on incoming SW? For example, are they shadows from the surrounding

topography that modifies incoming SW measured at the stations in early morning or late afternoon? Overall, if available, it would be interesting to know the topographic mask of each station with local horizon angles. Good question. These sites are very minimally affected by shading, although we have not determined the topographic mask for each station. The DEM is available with the dataset that would permit further assessment of this issue. We have added a brief comment to address this, "Sites were selected to minimize effects of topographic and vegetation shading, which only affect the sites briefly at very low sun angles."

P 6 L 17: it would be interesting to know at which height above the ground are typically measured wind speed and if snow depth is measured at all stations measuring wind speed. This information is useful to know at which height above the snow surface wind speed is measured in wintertime.

The wind speed and direction were measured ∼3m above the ground surface, as indicated in the published site metadata. We have added this information to the 1st sentence of the "Wind" section: "Wind speed (ws) and direction (wd) were continuously measured at seven sites at ∼3m above the ground surface." Snow depth and wind instrumentation were often, but not always, co-located (see Table 1). Thus, as snow depths averaged 5-70 cm at the locations with wind instrumentation, wind speed was measured ∼2.3-2.95 m above the snow surface in winter.

P 6 L 18: can the author add a comment about the representativeness of wind speed measurement? Does the surrounding vegetation influence wind speed measurement at some stations?

Although revised section 3.1 introduces the wind-sheltering observed at site 124b, we have added a sentence elaborating on this point on p. 6, lines 22-24: "Six of those sites are representative of surrounding wind conditions, and site 124b was deliberately established in a wind-sheltered aspen grove to better characterize snow accumulations in the upper portions of the basin."

P 6 L 25: are the raw precipitation data included in the dataset ? It would be interesting to have them if data users want to apply their own methods of correction following for example the recent SPICE project.

No, the raw precipitation is not available in the dataset. We have corrected and gap-filled the data as outlined in the paper. Readers who may be interested in the raw data may contact the ARS for the raw data if the quality-assured data here is insufficient for some reason.

P 8 L 2: based on Fig. 3, it appears that wind-induced snow transport strongly affects snow depth evolution at some stations. Could the authors comment more on the influence of wind-induced snow transport on the seasonal evolution of snow depth at this site? What are the stations that are typically exposed to wind-induced erosion and accumulation?

You are correct. This is primarily an issue at sites 124 and 124b, and is elaborated on page 7, lines 10-12: "Wind exposure at the upper measurement site 124 results in roughly the same corrected precipitation as at the lower elevation site 125. Precipitation catch at the sheltered site 124b is on average 1.2 times greater than at the wind-exposed site 124 (Table 1)."

P 8 L 4: can the author comment more on the method use to convert snow depth to SWE? Do they mean using the bulk snowpack density simulated by a snowpack scheme to convert measured snow depth into estimated SWE?

To clarify, we did not convert snow depth to SWE in this dataset; we only publish measured snow depths at JD. In this section of the manuscript, we simply pointed readers interested in determining SWE to a couple of papers that suggest methods for using snow density to model SWE from spatially-distributed snow depths. This is not intended to be an exhaustive review of that problem, which is outside the scope of this paper.

P 8 L 31: At P4 L3 the authors mention that the size of the catchment is 1.79 km2 which differs from 181 ha. The differences are small but what is the actual area of the catchment?

Thank you for pointing out this error. Based on the lidar data, the area was determined to be 181ha, and we have updated the reported area in the site description section.

P 9 L 15-16: it would interesting to mention in the conclusion that data are still collected at this experimental catchments and to precise whether the dataset will be updated on a regular basis to include the more recent years.

We have added a line to the conclusions to address this, "Data continue to be collected at the sites described here, and updated datasets will be published based on available resources."

Figure 1: terrain contour lines are hard to read.

See response to Figure 1 revision request above.

Figure 3 is very interesting and nicely shows the different temperature and precipitation conditions at for this catchment. However, the graphics showing snow depth evolution are hard to read. Maybe make two separate figures. The snow depth times series goes from October to late April whereas all the snow is generally gone in February. This graphics would be easier to read if they showed snow depth evolution from October to late February.

Thank you for this suggestion. We have expanded the x-axis, and corrected an alignment issue that had been introduced inadvertently into the previous version. Also see response to Figure 3 revision request above.

References Fierz, C. R. L. A., Armstrong, R. L., Durand, Y., Etchevers, P., Greene, E., McClung, D. M., ... & Sokratov, S. A. (2009). The international classification for seasonal snow on the ground (Vol. 25). Paris: UNESCO/IHP.

Please also note the supplement to this comment:
https://www.earth-syst-sci-data-discuss.net/essd-2017-112/essd-2017-112-AC1-
supplement.pdf
* * *